# Maternal and Offspring Cardiovascular Function following Pregnancy with Hypertensive Disorder

**DOI:** 10.3390/diagnostics13122007

**Published:** 2023-06-08

**Authors:** Coral Garcia-Gonzalez, Elena Nunez, Huijing Zhang, Kypros H. Nicolaides, Marietta Charakida

**Affiliations:** 1Harris Birthright Research Centre for Fetal Medicine, King’s College Hospital, London SE5 8BB, UK; coralgarciagonzalez@gmail.com (C.G.-G.); ele.nunez@hotmail.com (E.N.); huijing.zhang@nhs.net (H.Z.); charakidadoc@gmail.com (M.C.); 2School of Biomedical Engineering and Imaging Sciences, King’s College London, London SE1 7EH, UK

**Keywords:** maternal, offspring, cardiac function, preeclampsia, gestational hypertension, postpartum

## Abstract

Background: Hypertensive disorders of pregnancy (HDP) have been associated with increased cardiovascular risk for the mother and her offspring. However, it remains unknown whether cardiovascular changes are present in the postpartum period. Methods: This was a cross-sectional study of women with singleton pregnancies. We recruited 33 women (20 following preeclampsia and 13 following gestational hypertension) and an equal number of women with uncomplicated pregnancy. Conventional and more advanced echocardiographic modalities such as speckle tracking were used to assess maternal and offspring cardiac function at 3–9 months postpartum. Results: In women with HDP compared to those without, there was higher mean arterial pressure (mean 92.3 (SD 7.3) vs. 86.8 (8.3) mmHg, *p* = 0.007), left-ventricular mass indexed for body-surface area (64.5 (10.5) vs. 56.8 (10.03), *p* < 0.003), and E/e′ (3.6 (0.8) vs. 3.1 (0.9), *p* = 0.022). There were no significant differences between groups in maternal left-ventricular systolic-functional indices and in offspring cardiac function between groups. Conclusions: At 3–9 months postpartum, mothers with HDP had higher blood pressure, higher left-ventricular mass, and reduced left-ventricular diastolic function. However, in their offspring, cardiac function was preserved. These findings suggest that mothers who experienced an HDP would benefit from cardio-obstetric follow-up in the postpartum period.

## 1. Introduction

The development of hypertensive diseases of pregnancy (HDP), which includes gestational hypertension (GH) and preeclampsia (PE), occurs in approximately 5–8% of all pregnancies in developed countries [1]. The hallmark is the development of hypertension after 20 weeks of gestation with or without multiorgan involvement. The pathogenesis of these disorders remains poorly defined; however, placental hypoperfusion and hypoxia often exist. The conditions can trigger excessive systemic inflammatory response and have been associated with endothelial dysfunction and vasoconstriction, which can lead to systemic hypertension and end-organ hypoperfusion [1]. Although most of these complications seem to resolve with delivery, there are data to suggest that women who have experienced HDP may suffer from increased morbidity and mortality not only in the peripartum period but also during their lifetime [2,3]. For instance, large epidemiological studies have shown that women with HDP have double the risk of experiencing an adverse cardiovascular event and four times higher the risk of developing hypertension after birth compared to women who had a normotensive pregnancy [2,4]. However, from epidemiological studies it is difficult to ascertain whether these associations reflect progressive cardiac dysfunction that starts during pregnancy and continues thereafter or prolonged exposure to adverse underlying cardiovascular risk factors, which may even precede pregnancy [5]. For instance, our group and others demonstrated that women with HDP, compared to normotensive ones, had subclinical cardiac maladaptation during pregnancy [6,7,8]. In addition, it has been shown that some of the cardiac-functional changes persist in the peripartum period, but there is a paucity of information regarding cardiac changes that occur in the subsequent months [9,10].

Apart from the risk to the mother, HDP seems to impact the foetus, and a number of studies have documented that children who had been exposed to HDP have a higher risk of being hypertensive and obese [11,12]. Exposure to maternal PE has also been reported to be associated with a greater risk of stroke in adulthood, which may suggest that development of an adverse cardiovascular-risk-factor profile from adolescence might translate into cardiovascular events later in life [13]. A shared genetic risk between the mother and her offspring may be a contributor to increased offspring cardiovascular risk. However, offspring exposed to maternal PE had reduced endothelial dysfunction, which was not observed in their unexposed siblings, suggesting a direct intrauterine effect. Considering that the foetal myocardium is very sensitive to changes in afterload and hypoxia and given that increase in the number of myocyte plateaus during early life, one could argue that exposure to HDP may affect both offspring myocardial structure and function. This hypothesis is further supported by an epidemiological study in which adolescent offspring exposed to maternal PE had greater relative wall thickness and reduced left-ventricular end-diastolic volume, which could be early signs of concentric remodelling, as well as contribute to increased risk for cardiovascular-disease development [14].

In the current study, we performed detailed maternal and infant cardiac-functional assessment within the first year following delivery. The aim of the study was to characterize cardiovascular alterations that exist in both the mother and her child beyond the peripartum period following exposure to HDP.

## 2. Methods

### 2.1. Study Population

This is a cross-sectional study of women who experienced an HDP (gestational hypertension/preeclampsia) and an equal number of normotensive women of similar ethnic background who delivered at the same period at King’s College Hospital, London, UK.

Eligible women were recruited at birth and were asked to attend the Harris Birthright Research Institute for cardiovascular assessment at 3–9 months postpartum. Mothers were also requested to bring their child for cardiovascular evaluation. Inclusion criteria were delivery at term, singleton pregnancy, and absence of congenital heart defect. Mothers were excluded from participating in the study if they had breast implants, as these often compromise the echocardiographic acoustic windows. Patients were also not eligible to participate if they did not understand English and there was no available interpreter. All women provided written informed consent to participate in the Advanced Cardiovascular Imaging Study (REC No 18/NI/0013, 2018, IRAS ID: 237936).

### 2.2. Maternal and Childhood Characteristics

We recorded information on maternal age and racial origin (white, black, Asian, mixed). Weight and height were measured and body-mass index (kg/m^2^) was calculated at the clinical visit. Data on pregnancy outcome were collected from the hospital maternity records or the women’s general medical practitioners. Blood pressure was taken in the right arm using a Microlife automated device (3BTO-A2; Microlife, Taipei, Taiwan).

The obstetric records of all women were examined to determine the diagnosis of PE or GH. Diagnosis of PE and GH was made according to published criteria [15].

In all children, age in months was recorded and weight and height were measured. Information on feeding history and birthweight was obtained and the latter was converted to a z score according to gestational age of delivery.

### 2.3. Maternal and Childhood Cardiovascular Assessment

Cardiovascular assessment in our study population was performed using 2-dimensional and Doppler transthoracic echocardiography. Participants were asked to lie in the left lateral decubitus position to improve echocardiographic windows. Measurements were performed in the parasternal and apical views using a Canon Aplio i900 scanner (Canon Medical Systems Europe BV, Zoetermeer, The Netherlands) as per American Society of Echocardiography guidelines [16,17]. The systolic function of the left ventricle was assessed using M-mode in the parasternal long-axis view to calculate fraction shortening and the ejection fraction [18]. The values were indexed to the body surface area. Calculation of the relative wall thickness (RWT) was performed using the formula (2 × posterior wall thickness)/(LV internal diameter at end-diastole). According to the American Society of Echocardiography and the European Association of Cardiovascular Imaging guidelines, LV mass was categorized as either concentric (RWT > 0.42) or eccentric (RWT ≤ 0.42) hypertrophy [18]. Speckle-tracking analysis was performed in the four-chamber (Figure 1), two-chamber, and three-chamber views to calculate left-ventricular global-longitudinal systolic function. Increased negative values denote increased deformation and improved myocardial strain.

Left-ventricular diastolic function was assessed using pulse and tissue Doppler. The mitral inflow-velocity pattern was recorded from the apical four-chamber view. The mitral peak early (E) and late (A) diastolic-flow velocities were measured and the E/A ratio was calculated.

Pulsed-tissue Doppler recordings were obtained at the septal and lateral aspects of basal LV in the apical four-chamber view. Measurements obtained included peak systolic annular velocity (S), early mitral-annulus diastolic velocity (e′), and late diastolic velocity (a′). With the mitral inflow-velocity curve and the e′ velocity obtained from the septal and lateral sides of the mitral annulus, the E/e′ ratios and average E/e′ (septal and lateral) were also calculated.

The isovolumic-contraction (IVCT) and -relaxation times (IVRT) were measured from the tissue Doppler and the left-ventricular ejection time (ET) was measured as shown in Figure 2. The myocardial-performance index was calculated using the formula (IVCT + IVRT)/ET. The longitudinal function of the right ventricle was assessed by measuring the tricuspid-annular-plane systolic excursion in the four-chamber view, using M-mode across the lateral aspect of the tricuspid valve.

Hemodynamic measurements included assessment of cardiac output and peripheral vascular resistance. Cardiac output was assessed with echocardiography using the formula (stroke volume × heart rate). Stroke volume was derived from the left-ventricular-outflow-tract velocity–time integral. Peripheral vascular resistance was calculated as follows: (mean arterial pressure × 80)/cardiac output.

Analysis of all the cardiac measurements was performed by one fellow (CG) who was blinded to the participant characteristics and their diagnosis.

### 2.4. Statistical Analysis

Data were assessed for normality using histograms and quantile–quantile plots. Continuous variables were presented as the mean (standard deviation) and variables not following normal distribution as the median (interquartile range). Nominal variables were summarized as counts and percentages.

Comparison of cardiac measurements between the HDP group and controls was carried out using the T-test for normally distributed variables. For variables that were continuous but not normally distributed the Mann–Whitney U-test was used and for categorical variables the chi-squared test was used. General linear-regression models were used to assess the association between HDP and a range of echocardiographic parameters. Analysis was further adjusted for a pre-specified set of confounders, including age, race, mean arterial pressure, weight, and height.

Statistical analysis was performed with STATA package, version 15.1 (StataCorp, College Station, TX, USA). We deemed statistical significance to be at *p* < 0.05. All tests were two tailed.

## 3. Results

### 3.1. Maternal Characteristics

We studied 33 women with HDP (20 with PE and 13 with GH) and 33 women with normotensive pregnancy at around 6 months following delivery. None of the women were on antihypertensive medication at the time of assessment. Women in the HDP group, compared to controls, had higher weight, body-mass index, and mean arterial pressure (Table 1). There was no significant difference in heart rate and smoking history between the two groups.

### 3.2. Maternal Echocardiography

From the hemodynamic measurements, there was no significant difference in peripheral vascular resistance between the HDP group and controls. There was a trend towards increased cardiac output in women with HDP compared to those with normotensive pregnancy. From structural indices, women with HDP, compared to those with normotensive pregnancy, had higher LV mass indexed for body-surface area (Table 2). This association remained in multivariable analysis (coef. 5.7 [95th CI: 0.3, 11.0]). There was no evidence of concentric remodelling in women with HDP.

From diastolic left-ventricular functional indices in univariate analysis in women with HDP compared to controls, the E/A wave was reduced, whereas E/e′ was increased. However, in multivariable analysis, only E/e′ remained statistically significant (coef. 0.4 [95th CI: 0.03, 0.9]), suggesting a mild increase in left-ventricular-filling pressure in women with HDP compared to those with normotensive pregnancy.

As far as systolic and global functional indices were concerned, s′ was reduced in univariate analysis in women with HDP and there was no significant difference in myocardial-performance index between groups. Left-ventricular global longitudinal systolic strain was not statistically different between groups. In multivariable analysis, systolic-functional indices were not statistically different than controls.

### 3.3. Infant Characteristics and Echocardiography

There were no significant differences in weight, weight gain, type of feeding, or heart rate of infants of mothers with HDP compared to those with normotensive pregnancy. Infant length was marginally reduced in the HDP group compared to the controls (65.6 (3.7) vs. 68 (3.4), *p* = 0.012). There were no differences in birthweight z-score between groups. There were no significant differences in cardiac measurements in the infants of the HDP group compared to controls (Table 3).

## 4. Discussion

### 4.1. Main Findings

This study shows that women with HDP, compared to those without, at a mean interval of 6 months postpartum had increased left-ventricular mass and a mild reduction in left-ventricular diastolic function, but left-ventricular systolic function was not significantly different between the two groups. Although blood pressure was higher in women with HDP, this finding could not explain the noted cardiac alterations. All cardiac functional and structural changes were subclinical, and women were asymptomatic.

In contrast to the noted maternal cardiac changes in the HDP group, cardiovascular function in their offspring was not significantly different from that in the control group. These findings suggest that alterations in cardiac function and structure may contribute to the increased long-term cardiovascular risk of these women and support current guidelines that advocate monitoring women with HDP in the postpartum period. In contrast, our data did not reveal evidence of cardiac programming in infants who were exposed to maternal hypertension during pregnancy.

### 4.2. Comparison with Findings of Previous Studies and Interpretation of Results

A number of epidemiological studies have provided consistent and robust evidence that women following an HDP have an approximately four-fold increased risk of chronic hypertension [2]. The reported risk varies between the different types of HDP; with PE or GH in the early years after delivery and plateaus in subsequent years. For instance, in the recent registry of a Danish cohort study of 1.5 million women in which information on pregnancy complications and diagnosis of chronic hypertension was available, in the first 5 years after delivery women with GH compared with women with normotensive pregnancy had a 4–10 times higher risk of developing chronic hypertension, but this risk subsequently reduced to 2- to 2.5-fold [2]. A meta-analysis by Heida et al. showed that in women with PE the relative risk for development of chronic hypertension later in life was about 2.8, and this was similar for women who developed GH [19]. Several other studies examined the severity of HDP and the risk of developing chronic hypertension after pregnancy [20]. In a nationwide cohort study, Behrens et al. showed that women with severe PE had a 6.5 times higher risk of developing chronic hypertension one year after pregnancy than women with moderate PE [2]. In the current study, women with HDP had higher blood pressure compared to those with normotensive pregnancy, but none of them required antihypertensive therapy.

Apart from chronic hypertension, women with HDP are at increased risk for having other cardiovascular risk factors, including dyslipidemia and insulin resistance. A register-based cohort study from Denmark demonstrated a three-fold increased risk of type 2 diabetes for women with HDP compared with those with normotensive pregnancy [3]. However, apart from the accumulation of adverse cardiovascular risk factors, there are also data to support persistent cardiac remodelling and left-ventricular functional alterations for some women in the early postpartum period [20], whereas others showed recovery of cardiac indices at one year following delivery [21,22]. The degree of cardiac impairment appears to depend on the severity of PE and the timing of disease development, with early/preterm PE having worse cardiac impairment than term preeclampsia [23]. This finding aligns with results from epidemiological studies in which women with preterm PE had a 5 times higher relative risk of cardiovascular-disease development compared with women with term PE.

In the current study, we elected to study women beyond 3 months following delivery, aiming to avoid the peripartum period where pronounced haemodynamic changes occur in the mother, and tried to assess the presence of persistent cardiac alterations in the postpartum period. We showed, consistent with previous reports and a recently published study by Martino et al., that women with HDP had increased left-ventricular mass and increased left-ventricular-filling pressure, as demonstrated by the mild increase in the E/e′ index [20,24]. Cardiac changes in our study were mild and subclinical and remained after accounting for differences in the maternal risk-factor profile, including blood pressure. Considering that none of these women had had an echocardiogram prior to being pregnant, we are unable to determine whether these cardiac alterations reflect pre-existing cardiac differences or can be attributed solely to the development of HDP.

Previous studies have demonstrated that HDP is associated with foetal cardiac remodelling, i.e., globular hearts and reduced foetal myocardial function, and this pattern of foetal cardiac change was independent of the foetal growth pattern [25]. Both placental as well as hemodynamic parameters appeared as potential contributors to these alterations. In the current study, we had no information on foetal cardiac function during pregnancy but performed detailed cardiovascular assessment before the first year of life, aiming to identify persistent cardiac changes in infancy. The risk-factor profile was comparable between the HDP group and controls with no significant difference in birthweight, weight gain, and type of feeding. In addition, systolic and diastolic left-ventricular systolic functional indices were similar between the two groups, and there was no significant difference in left-ventricular mass or evidence of concentric remodelling. Our findings contradict the results of other studies where measurements were performed beyond infancy [12,14,26,27,28]. Fugelseth et al. reported that children at the age of 5–8 years who had been exposed to PE (*n* = 25) had smaller hearts compared to those with nonexposed ones, but the reference group included a large proportion of diabetic pregnancies [26]. Himmelmann et al., [27] in 52 children aged 10–16 years, showed no difference in the degree of increase in left-ventricular mass during adolescence in offspring of mothers with HDP compared to those without. Timpka et al., in 289 children aged 17.7 years, demonstrated greater relative wall thickness and reduced left-ventricular end-diastolic volume [14]. Lewandowski et al., using cardiac magnetic resonance, reported in 29 adults aged (20–39 years) who were exposed to maternal PE a decrease in the global longitudinal-peak systolic strain [28]. These discrepancies in the age of cardiovascular assessment make a comparisons between studies impossible. In addition, it remains unclear how much of the reported myocardial or vascular changes are driven by maternal hypertension before birth versus being a postnatal response.

### 4.3. Strengths and Limitations

Our study has several strengths. We performed a detailed cardiac phenotype in a homogeneous group of women and aimed to investigate the long-term cardiac consequences of HDP. We included women with term rather than preterm PE, as the former has higher incidence in the general population and is less responsive to interventions, i.e., aspirin during pregnancy [29]. Thus, our results might be applicable to the majority of women with HDP. In addition, we used as controls women who delivered at the same period to mothers with HDP, had a normotensive uncomplicated pregnancy, and had similar ethnic background to those with HDP. In this well-characterised group of women, we demonstrated that several months after delivery the maternal cardiovascular risk in women with HDP remains. However, no cardiovascular remodelling could be identified in their offspring, which suggests that accumulation of risk factors might be necessary to reveal any remodelling effect. Our study also has limitations. It is cross-sectional in nature; thus, we are unable to assess causal mechanisms and cannot provide any information about foetal and peripartum cardiovascular changes. In this population we did not perform hemodynamic measurements during pregnancy.

## 5. Conclusions

The national guidelines in the United Kingdom and recent American Heart guidelines recommend that after a diagnosis of PE, women should be counselled and followed up with for cardiovascular-risk-factor modification [30,31], however no conclusive evidence exists for an effective risk reduction strategy, and as such follow-up is probably not done in practice. In the current study, we demonstrated that mothers with HDP, compared to those without, have cardiac functional and structural alterations beyond the peripartum period, within the first six months post-delivery, which could not be accounted for by differences in blood pressure and other maternal characteristics. In contrast, cardiovascular function of the infant was unaffected to insults during pregnancy. Although the noted cardiac changes were mild and subclinical, our data would support current guidelines and suggest the need for maternal cardiovascular assessment in the postpartum period.

## Figures and Tables

**Figure 1 diagnostics-13-02007-f001:**
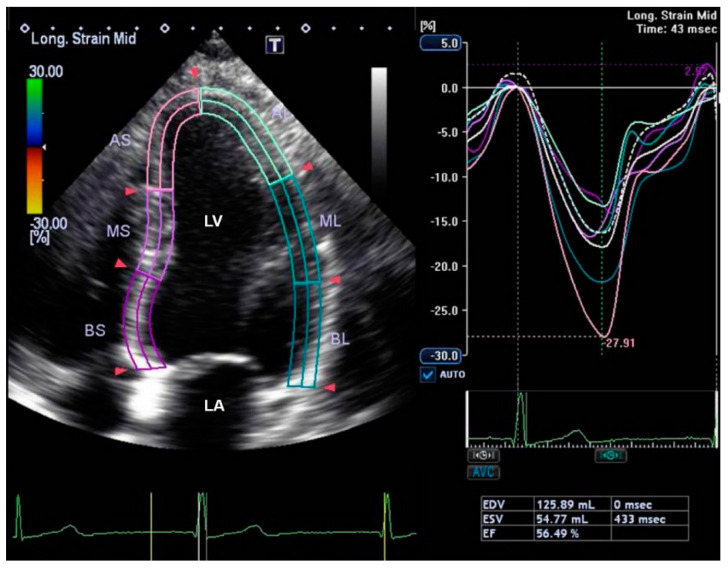
Illustration of the measurement of longitudinal systolic strain in the 4-chamber view of the heart. Global longitudinal strain is calculated automatically by having measurements also in the 2-chamber and 3-chamber view. Abbreviations: AS: anteroseptal, MS: mid-septal, BS: basal septal, AL: anterolateral, ML: mid-lateral, BL: basal lateral [18]. LA: left atrium, LV: left ventricle, EDV: end diastolic volume, ESV: end systolic volume, EF: ejection fraction.

**Figure 2 diagnostics-13-02007-f002:**
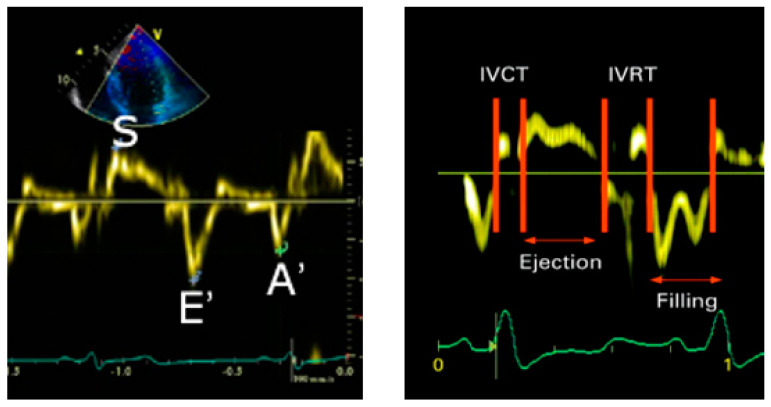
Illustration of tissue-Doppler imaging in the apical 4-chamber view. Analysis was performed for the peak systolic annular velocity (S), early mitral-annulus diastolic velocity (E′), and late diastolic velocity (A′). Timing intervals, isovolumic contraction time (IVCT), isovolumic relaxation time (IVRT), and ejection time (ET) were calculated as shown.

**Table 1 diagnostics-13-02007-t001:** Maternal characteristics.

	HDP (*n* = 33)	Controls (*n* = 33)	*p*-Value
Age (years)	32.7 (5.2)	32.2 (5.6)	0.751
Height (cm)	164.2 (6.9)	163.9 (7.3)	0.861
Weight (kg)	79.5 (11.9)	69.3 (14.6)	0.003
Body-mass index (kg/m^2^)	29.7 (4.8)	25.7 (5.20	0.002
Interval post delivery (days)	186 (49.8)	197 (29.1)	0.273
Mean arterial pressure (mmHg)	92.3 (7.3)	86.8 (8.3)	0.007
Heart rate (beats per minute)	72.4 (11.2)	68.4 (11.7)	0.161
Ethnicity			0.555
White	23 (69.7)	23 (69.7)	
Black	7 (21.2)	4 (12.1)	
Asian	3 (9.1)	4 (12.1)	
Smoking	1 (3.0)	1 (3.0)	0.356
Z birthweight	−0.67 (0.98)	−1.02(1.44)	0.249

Values given as mean (standard deviation) or *n* (%). HDP, hypertensive disorders of pregnancy.

**Table 2 diagnostics-13-02007-t002:** Maternal cardiac measurements.

Cardiac Index	HDP (*n* = 33)	Controls (*n* = 33)	*p*-Value
**Structural**			
Left ventricular mass indexed for body-surface area	64.5 (10.5)	56.8 (10.03)	0.003
Left ventricular relative wall thickness	0.37 (0.06)	0.40 (0.01)	0.078
**Haemodynamic**			
Cardiac output (L/min)	5.3 (1.3)	4.8 (1.0)	0.086
Peripheral-vascular resistance (dynes/cm^2^)	1423.8 (304.1)	1486.6 (323.3)	0.427
**Diastolic Indices**			
Mitral valve E (cm/s)	23.7 (3.8)	25.3 (3.4)	0.084
Mitral valve A (cm/s)	15.2 (3.1)	15.9 (3.6)	0.340
Mitral valve E/A	1.8 (0.6)	2.2 (0.9)	0.018
Mitral valve E/e′	3.6 (0.8)	3.1 (0.9)	0.022
**Systolic Indices**			
Myocardial-performance index	0.49 (0.1)	0.43 (0.1)	0.059
Mitral valve S	18.1 (2.8)	19.6 (3.0)	0.036
**Speckle Tracking**			
Left-ventricular global longitudinal strain (%)	−23.2 (2.7)	−23.9 (2.5)	0.240
Left-ventricular-ejection fraction (%)	62.6 (6.2)	62.9 (5.0)	0.853

Values given as mean (standard deviation). HDP, hypertensive disorders of pregnancy.

**Table 3 diagnostics-13-02007-t003:** Infant cardiac measurements.

Cardiac Index	HDP (*n* = 33)	Controls (*n* = 33)	*p*-Value
**Structural**			
Left-ventricular mass indexed for body-surface area	68.6 (31.1)	61.7 (18.5)	0.294
Left-ventricular relative wall thickness	0.4 (0.1)	0.3 (0.1)	0.071
**Haemodynamic**			
Cardiac output (L/min)	1.3 (0.2)	1.4 (0.3)	0.076
**Diastolic Indices**			
Mitral valve E (cm/s)	96.3 (15.5)	100.5 (15.7)	0.300
Mitral valve A (cm/s)	77.4 (19.7)	77.7 (15.3)	0.947
Mitral valve E/A	1.5 (0.5)	1.6 (0.5)	0.155
Mitral valve E/e′	4.6 (1.0)	4.5 (0.8)	0.614
**Systolic Indices**			
Myocardial-performance index	0.4 (0.1)	0.4 (0.1)	0.568
Tricuspid-annular-planar systolic excursion (mm)	14.1 (2.1)	14.8 (1.8)	0.213
Mitral valve S (cm/s)	13.9 (2.1)	14.4 (2.4)	0.338
**Speckle Tracking**			
Left-ventricular global-longitudinal-strain delta (%)	−21.3 (1.8)	−21.9 (2.1)	0.176
Left-ventricular-ejection fraction (%)	68.5 (7.8)	68.1 (7.3)	0.866

Values given as mean (standard deviation). HDP, hypertensive disorders of pregnancy.

## Data Availability

Anonymized data can be made available pending request and following permission from the Departmental Research and Development Office.

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
