# Peer review of "Maternal and Offspring Cardiovascular Function following Pregnancy with Hypertensive Disorder"

_diagnostics, 2023, doi:10.3390/diagnostics13122007_

Round 1

Reviewer 1 Report

I commend the Authors for their nice work and I believe the paper is already suitable for publication. The topic is of great interest and together with other recent studies on cardiovascular outcomes following HDP pregnancies it may change the way women suffering these disorders are followed-up after delivery.
The submitted paper is well written and organised, thus I have no concerns or suggestions to make.
Summary: Authors performed a cardiovascular assessment in 20 women following preeclampsia and 13 controls showing persistent sub-clinical cardiac abnormalities in women who suffered from Preeclampsia.

Author Response

Thank you so much for your positive feedback

Reviewer 2 Report

Congratulations to the authors to the very important study results given in the manuscript 'Maternal and offspring cardiovascular function following preg- 2 nancy with hypertensive disorder'.

To me this study is very clearly designed, presented in an easy understandable wording, and results and discussion with strength and limitations reflected in an objective manner.

Thanks for this very important contribution to refresh existing guidelines.

Author Response

Thank you so much for the positive feedback

Reviewer 3 Report

It is an original and very useful monitoring of risky cardiovascular disorders during pregnancy, especially in case of eclampsia. The report and the entire study are very useful for practitioners and are in line with the guidelines for the management of pregnant women with arterial hypertension.

It would be good to mention that additional guidelines should be followed in pregnant women with heart disease or a preformed heart by an established cardio-obstetric team

Author Response

Thank you we have modified our wording to say that these women would benefit from cardio-obstetric follow up in the postpartum period.